# OpenReview forum: "Are aligned neural networks adversarially aligned?"
_NeurIPS.cc/2023/Conference — NeurIPS 2023 poster_

### Official Review · Reviewer_n3yx · 2023-06-17

**Soundness:** 2 fair
**Presentation:** 1 poor
**Contribution:** 2 fair
**Rating:** 4
**Confidence:** 5

**Summary:**

This paper investigates the alignment of large language models in being helpful and harmless. While current NLP attacks are not powerful enough to consistently attack text models, adversarial input images can lead multimodal models to emit harmful content.

**Strengths:**

The paper focuses on studying the NLP attacks which might trigger adversarial behavior on aligned models. The research line seems novel.

**Weaknesses:**

1.	The paper's structure is disorganized, making it difficult for readers to follow. For example, the background section lacks a detailed explanation of existing attack methods and their comparison with the proposed approach. Additionally, there is repeated summarization of previous work throughout different sections, resulting in a confusing structure. Section 4.1 discusses prior attack methods, but important related works such as GBDA and ARCA are not introduced in Section 4.3.
2.	The expression in lines 171-175 is unclear, and it is difficult to comprehend the intended message. It would be helpful to provide a more detailed explanation of the two challenges mentioned. Why does it require |S| iterations of entire neural network forward passes? Why can't the sentence-level embeddings be directly inputted into the toxicity classifier for classification?
3.	The paper lacks sufficient details in explaining the experimental setup. For example, what does "StableVicuna" in Table 1 refer to? What value is chosen for K in line 216? Which vision model is used in line 319?
4.	The explanation of how the paper utilizes a brute force method in lines 252-257 needs further clarification. The description provided is confusing, and a more detailed explanation would be beneficial.
5.	TYPO: In line 258, "Tto" should be corrected to "To."

**Questions:**

still concern about the writing issues since the weakness part, more concerns here,
1.	Generally, it is not clear from the abstract what the contributions of this paper are. It is not until page 7 that the authors briefly explain what they are researching, and sections 1-5 contain a lot of writing that lacks meaning or is confusing.
2.	The introduction is difficult to understand as it is unclear what is being discussed. What are "continuous-domain images," and what is being conveyed in lines 56-62?
3.	What does the last sentence of background “we make ... often will not be” mean?
4.	What does the name of Sec3 “Threat Model” refer to?
5.	What is the “Attack Goal”? Line 152 – 158 do not bring any valuable information. Your definition is also confusing, and using substring detection for harmful text is not accurate because it is impossible to cover all possible harmful strings.
6.	I don't understand 171-172. If discrete tokens cannot be optimized, then how are all current language models trained? I don't know what 171-175 is referring, either.
7.	What is the “initial attack objective” in Line 182.
8.	Sec 5.1 and 5.2 appear to be your contributions, but I couldn't understand them, including your analysis in 6.3 and 6.4. They are too difficult to comprehend.

**Limitations:**

see the weakness and question parts.

---

> ### Author Rebuttal · Authors · 2023-08-10
>
> > The expression in lines 171-175 is unclear, and it is difficult to comprehend the intended message. It would be helpful to provide a more detailed explanation of the two challenges mentioned. Why does it require |S| iterations of entire neural network forward passes? Why can't the sentence-level embeddings be directly inputted into the toxicity classifier for classification?
>
> We will clarify this text to make it easier to follow. But because the adversary has to emit discrete tokens, they can't pass soft embeddings into a classifier if that's what's being asked here.
>
> > The paper lacks sufficient details in explaining the experimental setup. For example, what does "StableVicuna" in Table 1 refer to?
>
> StableVicuna refers to the StableVicuna model (https://stability.ai/blog/stablevicuna-open-source-rlhf-chatbot).
>
> > What value is chosen for K in line 216?
>
> We will update the paper to say that we choose K uniformly at random from [1, N).
>
> > Which vision model is used in line 319?
>
> As we say on Lines 321 and 326 we refer to Mini GPT-4 and LLaVA.
>
> > The explanation of how the paper utilizes a brute force method in lines 252-257 needs further clarification. The description provided is confusing, and a more detailed explanation would be beneficial.
>
> Lines 252-257 just summarize what Lucas et al. do for the image space. We explain what we do for our paper immediately following, on Lines 258-263. If this is still unclear we will attempt to improve the clarify of this aspect of the paper.

---

> > ### Comment · Area_Chair_9oS2 · 2023-08-19
> > **Thanks for your response**
> >
> > Dear authors,
> >
> > Thanks for your response. I think your explanation about the clarity and experimental details will help us evaluate your submission.
> >
> > Best,
> >
> > AC

---

### Official Review · Reviewer_5kqR · 2023-07-02

**Soundness:** 3 good
**Presentation:** 3 good
**Contribution:** 2 fair
**Rating:** 6
**Confidence:** 5

**Summary:**

The paper investigates the robustness of aligned neural networks, such as language models, by optimizing adversarial examples that aim to elicit unwanted behaviors (e.g., toxicity). The paper applies existing SOTA NLP adversarial attacks and finds they are insufficient to attack aligned models. However, the paper reports that the adversary's job is easier with multi-modal models due to the continuous space of image manipulation. Manipulating the image would cause the model to output unwanted and harmful content.

**Strengths:**

- The paper takes the first step towards synergizing adversarial ML research with the new attack vectors of chatbot and LLM security and alignment. I find this a great initiative to start a line of research to understand the similarities and differences between the two domains. The paper also constructs successful attacks against multimodal models. I find the paper generally well written (except for the need for some clarifications and more details). I appreciate the discussion that the absence of successful attacks should not be interpreted as proof that models are adversarially aligned.

**Weaknesses:**

- Despite the previous strengths, the paper does not show significant results as the examined attacks are not as successful compared to unaligned models (e.g., GPT-2). With the absence of any successful attacks, I am not very convinced with the paper's contribution and, taking the paper's initiatives aside, how much it can help advance research in this domain.

- Since the multimodal attacks show more potential, I would have appreciated deeper evaluation and discussion regarding this attack's potential. This would have increased the significance of the paper. For example, I am curious to know: how does the prompt (the user's query) impact the attack's success? How generalizable is the attack over different prompts given the same optimized image? How can the attack be stealthy (to include the threat model of malicious third-party data)? How would such attacks be different for other objectives (beyond toxicity)? How to produce more adversary-controlled attacks? I, of course, don't expect the paper to answer all these questions, but I believe more focus could have been put on evaluating these attacks.

- For reproducibility and better clarity, I think the experiments in section 6.2 need to have more details (to describe the attack optimization), possibly in an appendix.

- I was very confused by the experiment in sections 5.1 and 5.2, and I only started to get them after a couple of iterations. I was confused that 1) they are performed on GPT-2 (unaligned model), 2) the target token $q$ does not seem to be exclusively toxic, and 3) they don't follow the chatbot format explained in 4.1. I think this section needs to explain the motivation (as far as I understood, it shows the limitation of current attacks even on unaligned models and that it cannot find the prefix $p$ even though it always exists by construction).

- Minor comment:
line 151: I don't think the content filters are exclusive to the malicious third-party threat model. It seems that current systems (e.g., Bing) perform content filtering over users' input as well.

- Minor mistakes:
line 258: Tto
line 159: an language model
missing citation in the caption of table 3.


**Questions:**

- Why is the attacks' success rate defined by word overlap? Is it possible to use a classifier?

**Limitations:**

The paper addressed the ethical consideration and limitations.

---

> ### Author Rebuttal · Authors · 2023-08-10
>
> > Despite the previous strengths, the paper does not show significant results as the examined attacks are not as successful compared to unaligned models (e.g., GPT-2). With the absence of any successful attacks, I am not very convinced with the paper's contribution and, taking the paper's initiatives aside, how much it can help advance research in this domain.
>
> Prior work has found the attacking GPT-2 is much easier than attacking larger Vicuna or LLaMA models. If the prior NLP aren't even good enough for GPT-2, they will have no hope at succeeding on Vicuna/LLaMA. And so we do not perform this evaluation. But we can do this for the final paper if it would make the results more clear.
>
> > Since the multimodal attacks show more potential, I would have appreciated deeper evaluation and discussion regarding this attack's potential. This would have increased the significance of the paper.
>
> These are great questions that we can try to answer. But we have only limited space, and wanted our paper to be broad first and look at both NLP and vision-language models.  A few comments on a few of these follow.
>
> > how does the prompt (the user's query) impact the attack's success
>
> In preliminary experiments we found the user query does not impact the attack success rate. The adversarial image always works.
>
> > How can the attack be stealthy (to include the threat model of malicious third-party data)?
>
> We already do show that our attacks are stealthy and have very low L2 distortion (Table 3).
>
> > For reproducibility and better clarity, I think the experiments in section 6.2 need to have more details (to describe the attack optimization), possibly in an appendix.
>
> We wil l do this.
>
> > I was very confused by the experiment in sections 5.1 and 5.2, and I only started to get them after a couple of iterations. I was confused that 1) they are performed on GPT-2 (unaligned model), 2) the target token  does not seem to be exclusively toxic, and 3) they don't follow the chatbot format explained in 4.1. I think this section needs to explain the motivation (as far as I understood, it shows the limitation of current attacks even on unaligned models and that it cannot find the prefix even though it always exists by construction).
>
> We will try to clarify this setup, thank you for raising this. The idea here is that GPT-2 was previously found to be the easiest form of model to attack, and given that the attacks we're looking at couldn't even succeed at GPT-2, we don't even want to try them on the harder tasks. But we see how this is jarring in the current framing and will fix it to explain exactly why we do it this way and not what might be expected.

---

> > ### Comment · Reviewer_5kqR · 2023-08-14
> >
> > Thank you for your response.
> > I think the rebuttal answered most of my questions.
> >
> > I believe the paper has a good contribution and discusses an important topic. I don't expect it to provide a full investigation of attacks as this probably needs many follow-up investigations.
> >
> > My comment regarding the stealthiness was not about the image perturbation but about the text (e.g., attacks done by third-party data that manipulate the output), but this is probably beyond the scope of this current work.
> >
> > I strongly recommend that the final paper should try to have more clarifications on the setup and goals of the experiments.

---

### Official Review · Reviewer_NFJs · 2023-07-04

**Soundness:** 2 fair
**Presentation:** 1 poor
**Contribution:** 2 fair
**Rating:** 5
**Confidence:** 4

**Summary:**

**Summary:**
This paper discusses the alignment of neural networks, specifically language models, in the context of adversarial attacks. The authors explore the extent to which language models remain aligned even when interacting with adversarial users who construct worst-case inputs. They find that existing NLP-based adversarial attacks are insufficiently powerful to reliably attack aligned text models, but multimodal models can be caused to emit hateful and toxic content through adversarial perturbation of the input image. The authors hope that stronger NLP attacks will enable comprehensive robustness evaluations of aligned LLMs, which can help inform alignment researchers to develop improved model alignment techniques that remain reliable in adversarial environments.



**Pros:**

1.The paper addresses an important and timely topic in the field of AI, which is the adversarial robustness of language models.

2.The authors provide a clear and concise explanation of the problem and their approach to studying it.

3.The paper presents empirical evidence to support their findings, which can be useful for other researchers in the field.


**Cons:**

[Concept misleading] The concepts overlap between harmness and adversary. Adversarial attack is a set of algorithms that adds adversarial perturbation to sentences cumulatively based on some discrete optimization techniques. While harmness is a property that reflects the generation quality in terms of ethics.

[Insufficient survey] The authors choose the title focusing on adversarial attack, they’re actually talking about preventing harmness instead of general adversarial attack, so the title is too wide.
There’re lots of defenses and attacks in the NLP field, which has not been discussed by the authors, only mentioned in related work, and they didn’t discuss the relevance of such works.

[Limited novelty] This paper mainly discusses how to hack language models to produce harmful contents without their own designed methods. Overall, it’s an empirical analysis with limited baselines, models and benchmarks.

[Writing issue] The authors use too much space to discuss the background on adversarial attack, over two pages. In fact, using one simple paragraph can cover all the basics in an adversarial attack. Besides, some parts of the paper are not necessary, for example, the Multimodal text-vision models in related works  can be omitted. The focus of this paper is how to hack LLM to produce harmful contents, the text-vision model is only one of the base models in the experiments, it’s not in the paper’s focus.

**Strengths:**

**Pros:**

1.The paper addresses an important and timely topic in the field of AI, which is the adversarial robustness of language models.

2.The authors provide a clear and concise explanation of the problem and their approach to studying it.

3.The paper presents empirical evidence to support their findings, which can be useful for other researchers in the field.

**Weaknesses:**

**Cons:**

[Concept misleading] The concepts overlap between harmness and adversary. Adversarial attack is a set of algorithms that adds adversarial perturbation to sentences cumulatively based on some discrete optimization techniques. While harmness is a property that reflects the generation quality in terms of ethics.

[Insufficient survey] The authors choose the title focusing on adversarial attack, they’re actually talking about preventing harmness instead of general adversarial attack, so the title is too wide. There’re lots of defenses and attacks in the NLP field, which has not been discussed by the authors, only mentioned in related work, and they didn’t discuss the relevance of such works.

[Limited novelty] This paper mainly discusses how to hack language models to produce harmful contents without their own designed methods. Overall, it’s an empirical analysis with limited baselines, models and benchmarks.

[Writing issue] The authors use too much space to discuss the background on adversarial attack, over two pages. In fact, using one simple paragraph can cover all the basics in an adversarial attack. Besides, some parts of the paper are not necessary, for example, the Multimodal text-vision models in related works can be omitted. The focus of this paper is how to hack LLM to produce harmful contents, the text-vision model is only one of the base models in the experiments, it’s not in the paper’s focus.

**Questions:**

Refer to Weaknesses

**Limitations:**

Yes

---

> ### Author Rebuttal · Authors · 2023-08-10
>
> > [Concept misleading] The concepts overlap between harmness and adversary. Adversarial attack is a set of algorithms that adds adversarial perturbation to sentences cumulatively based on some discrete optimization techniques. While harmness is a property that reflects the generation quality in terms of ethics.
>
> We do not understand this comment. Adversarial examples are inputs designed by an adversary to cause a model to behave incorrectly. The "correct" behavior of the system is to not swear at the user. We show how to make the system swear at the User. Thus it is an adversarial example.
>
> > [Insufficient survey] The authors choose the title focusing on adversarial attack, they’re actually talking about preventing harmness instead of general adversarial attack, so the title is too wide.
>
> We are generating adversarial examples; see above.
>
> > [Limited novelty] This paper mainly discusses how to hack language models to produce harmful contents without their own designed methods. Overall, it’s an empirical analysis with limited baselines, models and benchmarks.
>
> We are the first paper to be able to show a new vulnerability, and so there is no baseline to compare against. We make our own baselines as described in the paper.
>
> > The focus of this paper is how to hack LLM to produce harmful contents, the text-vision model is only one of the base models in the experiments, it’s not in the paper’s focus.
>
> We respectfully disagree. The focus of our paper is to understand if machine learning models are aligned. The first half of our results focus on the language-only setting, where we show existing NLP attacks are not currently good enough to resolve the question we ask. Thus we encourage future work here. The second half of our paper shows that vision attacks are good enough to demonstrate these models are not adversarially aligned.

---

> > ### Comment · Area_Chair_9oS2 · 2023-08-19
> > **Thanks for your response**
> >
> > Dear authors,
> >
> > Thanks for your detailed response. I think your explanation about concept misleading, insufficient survey and other questions will help us evaluate your submission.
> >
> > Best,
> >
> > AC

---

> > ### Comment · Reviewer_NFJs · 2023-08-20
> >
> > Dear authors,
> >
> > Thanks for your response! After reading some related work, I found several of your points to be valid. Consequently, I've increased my score to 5.

---

### Official Review · Reviewer_DTQ2 · 2023-07-07

**Soundness:** 3 good
**Presentation:** 3 good
**Contribution:** 3 good
**Rating:** 6
**Confidence:** 4

**Summary:**

This work evaluates whether aligned neural networks are adversarially aligned, where in this case “adversarially aligned” means that they do not output toxic content. They analyze two classes of models: LLMs using the chat interface, and multimodal models that answer questions given text and images. They find that existing NLP attacks can often fail to find sequences of tokens that produce toxic text on LLMs trained for chat, but attacks on the image inputs to multimodal models work every time, with minimal l_2 distortion. One possible reason for the relatively low performance of these attacks is that the models are actually aligned. To test this, the authors measure the performance of NLP attacks on examples where they know that a solution exists, and find that on these, the best existing attack (ARCA) only produces solutions between 13 and 90% of the time.


**Strengths:**

* This paper introduces an important way of evaluating the alignment of language models (adversarial alignment), which largely makes sense and seems much more concrete than many existing ways of measuring alignment.
* The paper’s results on multimodal models highlight severe weaknesses in these systems, with ramifications for how they should be deployed.
* The paper has some reasonable benchmarking of existing NLP attacks. I especially liked the evaluation where the attack is known to exist, and feel like the authors did a good job showing that the attacks they tested have significant room for improvement.


**Weaknesses:**

* The paper is more of an evaluation paper (in the sense that they set up a threat model, then run existing attacks on it), but only runs two NLP attacks (while in reality there exist many, e.g. word substitution attacks like [1]). The paper also only considers toxicity as a source of misalignment, while deception, etc. might be other metrics. For an evaluation / threat model definition paper, it would be nice to see more diverse benchmarking.
* The authors claim that their NLP threat model (inserting an agent token before generating response) is much more realistic than the setting in Jones et al. (where no agent token exists), so following the latter would be “putting words in the LLM’s mouth and then forcing it to continue from there—something that is an unfair evaluation” [204-205]. However, this attack setup seems reasonable for many other systems (e.g., autocomplete), which should still be adversarially aligned; using the current threat model alone may give a false sense of security.
* There are some missing experimental details; the authors say they run the ARCA attack and GBDA attack, but don’t specify hyperparameters (e.g., learning rate for GBDA, number of averaged gradients for ARCA, number of runs of each attack), which harms reproducibility. It might help to also define the attacks used.

[1] Alzantot et al., Generating natural language adversarial examples EMNLP, 2018.



**Questions:**

* In Table 2, is there also guaranteed to be a solution for 2, 5, and 10 tokens? Or just the optimal, and the brute force attack managed to find it.
* [292 - 293]: “ARCA is significantly stronger than GBDA (consistent with prior work) but even ARCA passes less than half of the time given the optimal number of tokens.”; in Table 2 it seems like ARCA passes 53% of the time?
* [210 - 211]: “we construct a dataset of potentially toxic messages”. Do you construct the dataset? It sounds like it comes from the ARCA paper?


**Limitations:**

Yes.

---

> ### Author Rebuttal · Authors · 2023-08-10
>
> > But only runs two NLP attacks (while in reality there exist many, e.g. word substitution attacks like [1]).
>
> We choose the two strongest attacks from the literature which dominate prior work. We can add additional entries for the final paper but because prior work already found these two attacks (and ARCA especially) to be the most effective, we do not expect other, older, attacks to work better.
>
> > However, this attack setup seems reasonable for many other systems (e.g., autocomplete)
>
> We agree, but there do not currently exist any autocomplete systems that have been RLHF trained. And so it is only meaningful to attack chat-RLHF'd systems, which require this new prompting strategy.
>
> > There are some missing experimental details; the authors say they run the ARCA attack and GBDA attack, but don’t specify hyperparameters
>
> We run all attacks with default hyperparameters as suggested in the paper and code. We will add them to the appendix. We additionally tried experiments running these attacks 10x longer but did not see significant success rate improvements.

---

> > ### Comment · Area_Chair_9oS2 · 2023-08-19
> > **Thanks for your response**
> >
> > Dear authors,
> >
> > Thanks for your response. I think your response has addressed some of the reviewer's concerns.
> >
> > Best,
> >
> > AC

---

### Comment · Area_Chair_9oS2 · 2023-08-13
**Thanks for your insightful reviews**

Dear reviewers,

Thanks for your insightful reviews. It appears that we have received a little bit of divergent reviews. Now that the authors have provided a response for each review, we kindly request you take a moment to review these texts and determine if they effectively address the concerns you raised.

AC

---

### Decision · Program_Chairs · 2023-09-21

**Decision:**

Accept (poster)

**Comment:**

The paper has addressed an interesting topic—how to evaluate the alignment of language models. The authors provide a clear and concise explanation of the problem and their approach. Experimental results show that existing NLP-based adversarial attacks are insufficiently powerful to reliably attack aligned text models. However, multimodal models can be made to emit hateful and toxic content through adversarial perturbation of the input image.

We received slightly varying reviews, with most concerns regarding the writing. Subsequently, the authors provided succinct responses to each review, resulting in higher scores.